# Multilabel reductions: what is my loss optimising?

**Aditya Krishna Menon, Ankit Singh Rawat, Sashank J. Reddi, and Sanjiv Kumar**
Google Research
New York, NY 10011
{adityakmenon, sashank, ankitsrawat, sanjivk}@google.com

## Abstract

Multilabel classification is a challenging problem arising in applications ranging from information retrieval to image tagging. A popular approach to this problem is to employ a reduction to a suitable series of binary or multiclass problems (e.g., computing a softmax based cross-entropy over the relevant labels). While such methods have seen empirical success, less is understood about how well they approximate two fundamental performance measures: *precision@k* and *recall@k*. In this paper, we study five commonly used reductions, including the one-versus-all reduction, a reduction to multiclass classification, and *normalised* versions of the same, wherein the contribution of each instance is normalised by the number of relevant labels. Our main result is a formal justification of each reduction: we explicate their underlying risks, and show they are each consistent with respect to *either* precision or recall. Further, we show that in general *no* reduction can be optimal for *both* measures. We empirically validate our results, demonstrating scenarios where normalised reductions yield recall gains over unnormalised counterparts.

## 1 Introduction

Multilabel classification is the problem of predicting *multiple* labels for a given instance [Tsoumakas and Katakis, 2007, Zhang and Zhou, 2014]. For example, in information retrieval, one may predict whether a number of documents are relevant to a given query [Manning et al., 2008]; in image tagging, one may predict whether several individuals' faces are contained in an image [Xiao et al., 2010]. In the *extreme classification* scenario where the number of potential labels is large, naïve modelling of all possible label combinations is prohibitive. This has motivated a number of algorithms targeting this setting [Agrawal et al., 2013, Yu et al., 2014, Bhatia et al., 2015, Jain et al., 2016, Babbar and Schölkopf, 2017, Yen et al., 2017, Prabhu et al., 2018, Jain et al., 2019, Reddi et al., 2019]. One popular strategy is to employ a reduction to a series of binary or multiclass problems, and thereafter treat labels independently. Such reductions significantly reduce the complexity of learning, and have seen empirical success. For example, in the *one-versus-all* reduction, one reduces the problem to a series of independent binary classification tasks [Brinker et al., 2006, Dembczyński et al., 2010, 2012]. Similarly, in the *pick-one-label* reduction, one reduces the problem to multiclass classification with a randomly drawn positive label [Boutell et al., 2004, Jernite et al., 2017, Joulin et al., 2017].

The theoretical aspects of these reductions are less clear, however. In particular, precisely what properties of the original multilabel problem do these reductions preserve? While similar questions are well-studied for reductions of multiclass to binary classification [Zhang, 2004, Tewari and Bartlett, 2007, Ramaswamy et al., 2014], reductions for multilabel problems have received less attention, despite their wide use in practice. Recently, Wydmuch et al. [2018] established that the pick-one-label is *inconsistent* with respect to the precision@$k$, a key measure of retrieval performance. On the other hand, they showed that a probabilistic label tree-based implementation of one-versus-all is consistent. This intriguing observation raises two natural questions: what can be said about consistency of *other* reductions? And does the picture change if we consider a different metric, such as the *recall@k*?

| Reduction | Notation | Bayes-optimal $f_i^*(x)$ | Prec@$k$ | Rec@$k$ | Consistency analysis |
|---|---|---|---|---|---|
| One-versus-all | $\ell_{\mathrm{OVA}}$ | $\mathbb{P}(y_i = 1 \mid x)$ | ✓ | ✗ | [Wydmuch et al., 2018] |
| Pick-all-labels | $\ell_{\mathrm{PAL}}$ | $\mathbb{P}(y_i = 1 \mid x)/N(x)$ | ✓ | ✗ | This paper |
| One-versus-all normalised | $\ell_{\mathrm{OVA-N}}$ | $\mathbb{P}(y_i' = 1 \mid x)$ | ✗ | ✓ | This paper |
| Pick-all-labels normalised | $\ell_{\mathrm{PAL-N}}$ | $\mathbb{P}(y_i' = 1 \mid x)$ | ✗ | ✓ | This paper |
| Pick-one-label | $\ell_{\mathrm{POL}}$ | $\mathbb{P}(y_i' = 1 \mid x)$ | ✗ | ✓ | This paper |

Table 1: Summary of reductions of multilabel to binary or multiclass classification studied in §4.1 of this paper. For each reduction, we specify the notation used for the loss function; the Bayes-optimal scorer for the $i$th label, assuming random (instance, label) pairs $(x, y)$; whether the reduction is consistent for to the multilabel precision@$k$ and recall@$k$; and where this analysis is provided. Here, $\mathbb{P}(y_i = 1 \mid x)$ denotes the marginal probability of the $i$th label; $\mathbb{P}(y_i' = 1 \mid x)$ denotes a nonlinear transformation of this probability, per (5); and $N(x)$ is the expected number of relevant labels for $x$, also per (12).

In this paper, we provide a systematic study of these questions by investigating the consistency of five multilabel reductions with respect to both precision@$k$ and recall@$k$:

(a) the one-versus-all reduction (OVA) to independent binary problems [Dembczyński et al., 2010];
(b) a pick-all-labels (PAL) multiclass reduction, wherein a separate multiclass example is created for each positive label [Reddi et al., 2019];
(c) a *normalised* one-versus-all reduction (OVA-N), where the contribution of each instance is normalised by the number of relevant labels;
(d) a *normalised* pick-all-labels reduction (PAL-N), with the same normalisation as above; and,
(e) the pick-one-label (POL) multiclass reduction, wherein a single multiclass example is created by randomly sampling a positive label [Joulin et al., 2017, Jernite et al., 2017].

Our main result is a formal justification of each reduction: we explicate the multilabel risks underpinning each of them (Proposition 5), and use this to show they are each consistent with respect to *either* precision or recall (Proposition 9, 10). Further, this dichotomy is inescapable: the Bayes-optimal scorers for the two measures are fundamentally incompatible (Corollary 2, 4), except in trivial cases where all labels are conditionally independent, or the number of relevant labels is constant.

This finding has two important implications. First, while the above reductions appear superficially similar, they target fundamentally different performance measures. Consequently, when recall is of primary interest, such as when there are only a few relevant labels for an instance [Lapin et al., 2018], employing the "wrong" reduction can potentially lead to suboptimal performance. Second, the probability scores obtained from each reduction must be interpreted with caution: except for the one-versus-all reduction, they do *not* coincide with the marginal probabilities for each label. Naïvely using these probabilities for downstream decision-making may thus be sub-optimal.

In summary, our contributions are the following (see also Table 1):

(1) we formalise the implicit multilabel loss and risk underpinning five distinct multilabel learning reductions (§4.1) to a suitable binary or multiclass problem (Proposition 5).
(2) we establish suitable consistency of each reduction: the *unnormalised* reductions are consistent for precision, while the *normalised* reductions are consistent for recall (Proposition 9, 10).
(3) we empirically confirm that normalised reductions can yield recall gains over unnormalised counterparts, while the latter can yield precision gains over the former.

## 2 Background and notation

We formalise the multilabel classification problem, and its special case of multiclass classification.

### 2.1 Multilabel classification

Suppose we have an instance space $\mathcal{X}$ (e.g., queries) and label space $\mathcal{Y} \doteq \{0, 1\}^L$ (e.g., documents) for some $L \in \mathbb{N}_+$. Here, $L$ represents the total number of possible labels. Given an instance $x \in \mathcal{X}$ with label vector $y \in \mathcal{Y}$, we interpret $y_i = 1$ to mean that the label $i$ is "relevant" to the instance $x$. Importantly, there may be multiple relevant labels for a given instance. Our goal is, informally, to find a ranking over labels given an instance (e.g., rank the most relevant documents for a query).

More precisely, let $\mathbb{P}$ be a distribution over $\mathcal{X} \times \mathcal{Y}$, where $\mathbb{P}(y \mid x)$ denotes the suitability of label vector $y$ for instance $x$, and $\mathbb{P}(\mathbf{0}_L \mid x) = 0$ (i.e., each instance must have at least one relevant

label).[1] Our goal is to learn a *scorer* $f \colon \mathcal{X} \to \mathbb{R}^L$ that orders labels according to their suitability (e.g., scores documents based on their relevance for a given query). We evaluate a scorer according to the *precision-at-k* and *recall-at-k* for given $k \in [L] \doteq \{1, 2, \ldots, L\}$ [Lapin et al., 2018]:[2]

$$\mathrm{Prec}@k(f) \doteq \mathop{\mathbb{E}}_{(x,y)}\left[\frac{|\mathrm{rel}(y) \cap \mathrm{Top}_k(f(x))|}{k}\right] \qquad \mathrm{Rec}@k(f) \doteq \mathop{\mathbb{E}}_{(x,y)}\left[\frac{|\mathrm{rel}(y) \cap \mathrm{Top}_k(f(x))|}{|\mathrm{rel}(y)|}\right],$$
(1)

where $\mathrm{Top}_k(f)$ returns the top $k$ scoring labels according to $f$ (assuming no ties), and $\mathrm{rel}(y)$ denotes the indices of the relevant (positive) labels of $y$. In a retrieval context, the recall@$k$ may be favourable when $k \gg \mathrm{rel}(y)$ (i.e., we retrieve a large number of documents, but there are only a few relevant documents for a query), since the precision will degrade as $k$ increases [Lapin et al., 2018].

Optimising either of these measures directly is intractable, and so typically one picks a *multilabel surrogate* loss $\ell_{\mathrm{ML}} \colon \{0,1\}^L \times \mathbb{R}^L \to \mathbb{R}_+$, and minimises the *multilabel risk* $R_{\mathrm{ML}}(f) \doteq \mathbb{E}_{(x,y)}[\ell_{\mathrm{ML}}(y, f(x))]$. A *Bayes-optimal* scorer $f^*$ is any minimiser of $R_{\mathrm{ML}}$. For several performance measures, $f^*$ is any monotone transformation of the *marginal label probabilities* $\mathbb{P}(y_i = 1 \mid x)$ [Dembczyński et al., 2010, Koyejo et al., 2015, Wu and Zhou, 2017]. Thus, accurate estimation of these marginals suffices for good performance on these measures. This gives credence to the existence of efficient reductions that preserve multilabel classification performance, a topic we shall study in §4.

## 2.2 Multiclass classification

Multiclass classification is a special case of multilabel classification where each instance has only one relevant label. Concretely, our label space is now $\mathcal{Z} \doteq [L]$. Suppose there is an unknown distribution $\mathbb{P}$ over $\mathcal{X} \times \mathcal{Z}$, where $\mathbb{P}(z \mid x)$ denotes the suitability of label $z$ for instance $x$. We may now evaluate a candidate scorer $f \colon \mathcal{X} \to \mathbb{R}^L$ according to the *top-k* risk for given $k \in [L]$:

$$R_{\mathrm{top}-k}(f) \doteq \mathbb{P}(z \notin \mathrm{Top}_k(f(x))).$$
(2)

This measure is natural when we can make $k$ guesses as to an instance's label, and are only penalised if all guesses are incorrect [Lapin et al., 2018]. This is equivalent to the expected *top-k loss*, given by

$$\ell_{\mathrm{top}-k}(z, f) \doteq [\![z \notin \mathrm{Top}_k(f)]\!].$$
(3)

When $k = 1$, we obtain the zero-one or misclassification loss. For computational tractability, rather than minimise (2) directly, one often optimises a surrogate loss $\ell_{\mathrm{MC}} \colon [L] \times \mathbb{R}^L \to \mathbb{R}_+$, with risk $R_{\mathrm{MC}}(f) \doteq \mathbb{E}_{(x,z)}[\ell_{\mathrm{MC}}(z, f(x))]$. Several surrogate losses have been studied [Zhang, 2004, Ávila Pires and Szepesvári, 2016], the most popular being the softmax cross-entropy $\ell_{\mathrm{SM}}(i, f) \doteq -f_i + \log \sum_{j \in [L]} e^{f_j}$, with $f_i$ being the $i$th coordinate of the vector $f$. *Consistency* of such surrogates with respect to the top-$k$ error have been considered in several recent works (see e.g., [Lapin et al., 2015, Lapin et al., 2018, Yang and Koyejo, 2019]). We say that $\ell_{\mathrm{MC}}$ is *consistent* for the top-$k$ error if driving the excess risk for $\ell_{\mathrm{MC}}$ to zero also drives the excess risk for the $\ell_{\mathrm{top}-k}$ to zero; that is, for any sequence $(f_n)_{n=1}^{\infty}$ of scorers,

$$\mathrm{reg}(f_n; \ell_{\mathrm{MC}}) \to 0 \implies \mathrm{reg}(f_n; \ell_{\mathrm{top}-k}) \to 0,$$
(4)

where the *regret* of a scorer $f$ with respect to $\ell_{\mathrm{MC}}$ is $\mathrm{reg}(f; \ell_{\mathrm{MC}}) \doteq R_{\mathrm{MC}}(f) - \inf_{g \colon \mathcal{X} \to \mathbb{R}^L} R_{\mathrm{MC}}(g)$.

# 3 Optimal scorers for multilabel precision and recall@$k$

Our focus in this paper is on multilabel classification performance according to the precision@$k$ and recall@$k$ (cf. (1)). It is therefore prudent to ask: what are the Bayes-optimal predictions for each measure? Answering this gives insight into what aspects of a scorer these measures focus on.

Recently, Wydmuch et al. [2018] studied this question for precision@$k$, establishing that it is optimised by any order-preserving transformation of the marginal probabilities $\mathbb{P}(y_i = 1 \mid x)$.

**Lemma 1** ([Wydmuch et al., 2018]). *The multilabel precision@$k$ of a scorer $f \colon \mathcal{X} \to \mathbb{R}^L$ is*

$$\mathrm{Prec}@k(f) = \mathop{\mathbb{E}}_{x}\left[\sum_{i \in \mathrm{Top}_k(f(x))} \frac{1}{k} \cdot \mathbb{P}(y_i = 1 \mid x)\right].$$

**Corollary 2** ([Wydmuch et al., 2018]). *Assuming there are no ties in the probabilities* $\mathbb{P}(y_i = 1 \mid x)$,

$$f^* \in \operatorname*{argmax}_{f \colon \mathcal{X} \to \mathbb{R}} \operatorname{Prec@}k(f) \iff (\forall x \in \mathcal{X}) \operatorname{Top}_k(f^*(x)) = \operatorname{Top}_k\left([\mathbb{P}(y_i = 1 \mid x)]_{i=1}^L\right).$$

We now show that, by contrast, the recall@$k$ will in general *not* encourage ordering by the marginal probabilities. Indeed, it can be expressed in a nearly identical form to the precison@$k$, but with a crucial difference: the marginal probabilities are transformed by an additional nonlinear weighting.

**Lemma 3.** *The multilabel recall@$k$ of a scorer $f \colon \mathcal{X} \to \mathbb{R}^L$ is*

$$\operatorname{Rec@}k(f) = \mathbb{E}_x\left[\sum_{i \in \operatorname{Top}_k(f(x))} \mathbb{P}(y_i' = 1 \mid x)\right]$$

$$\mathbb{P}(y_i' = 1 \mid x) \doteq \mathbb{P}(y_i = 1 \mid x) \cdot \mathbb{E}_{y_{\neg i} \mid x, y_i = 1}\left[\frac{1}{1 + \sum_{j \neq i} y_j}\right], \tag{5}$$

*where $y_{\neg i}$ denotes the vector of all but the $i$th label, i.e., $(y_1, \ldots, y_{i-1}, y_{i+1}, \ldots, y_L) \in \{0, 1\}^{L-1}$.*

The "transformed" probabilities $\mathbb{P}(y_i' = 1 \mid x)$ will in general not preserve the ordering of the marginal probabilities $\mathbb{P}(y_i = 1 \mid x)$, owing to the multiplication by a non-constant term. We thus have the following, which is implicit in Wydmuch et al. [2018, Proposition 1], wherein it was shown the pick-one-label reduction is *in*consistent with respect to precision.

**Corollary 4.** *Assuming there are no ties in the probabilities* $\mathbb{P}(y_i' = 1 \mid x)$,

$$f^* \in \operatorname*{argmax}_{f \colon \mathcal{X} \to \mathbb{R}} \operatorname{Rec@}k(f) \iff (\forall x \in \mathcal{X}) \operatorname{Top}_k(f^*(x)) = \operatorname{Top}_k\left([\mathbb{P}(y_i' = 1 \mid x)]_{i=1}^L\right).$$

*Further, the order of $\mathbb{P}(y_i' = 1 \mid x)$ and $\mathbb{P}(y_i = 1 \mid x)$ do not coincide in general.*

One implication of Corollary 4 is that, when designing reductions for multilabel learning, one must carefully assess which of these two measures (if any) the reduction is optimal for. We cannot hope for a reduction to be optimal for *both*, since their Bayes-optimal scorers are generally incompatible. We remark however that one special case where $\mathbb{P}(y_i' = 1 \mid x) = \mathbb{P}(y_i = 1 \mid x)$ is when $\sum_{i \in [L]} y_i$ is a constant for every instance, which may happen if the labels are conditionally independent.

Having obtained a handle on these performance measures, we proceed with our central object of inquiry: are they well approximated by existing multilabel reductions?

# 4 Reductions from multi- to single-label classification

We study five distinct reductions of multilabel to binary or multiclass classification. Our particular interest is in what multilabel performance measure (if any) these reductions implicitly aim to optimise. As a first step to answering this, we explicate the multilabel loss and risk underpinning each of them.

## 4.1 Multilabel reductions: loss functions

Recall from §2.1 that a standard approach to multilabel classification is minimising a suitable multilabel loss function $\ell_{\mathrm{ML}} \colon \{0, 1\}^L \times \mathbb{R}^L \to \mathbb{R}_+$. To construct such a loss, a popular approach is to *decompose* it into a suitable combination of binary or multiclass losses; implicitly, this is a *reduction* of multilabel learning to a suitable binary or multiclass problem.

We consider five distinct decompositions (see Table 2). For each, we explicate their underlying multilabel loss in order to compare them on an equal footing. Despite the widespread use of these reductions, to our knowledge, they have not been explicitly compared in this manner by prior work. Our goal is to thus provide a unified analysis of distinct methods, similar to the analysis in Dembczyński et al. [2012] of the label dependence assumptions underpinning multilabel algorithms.

**One-versus-all** (OVA). The first approach is arguably the simplest: we train $L$ independent binary classification models to predict each $y_i \in \{0, 1\}$. This can be interpreted as using the multilabel loss

$$\ell_{\mathrm{OVA}}(y, f) \doteq \sum_{i \in [L]} \ell_{\mathrm{BC}}(y_i, f_i) = \sum_{i \in [L]} \{y_i \cdot \ell_{\mathrm{BC}}(1, f_i) + (1 - y_i) \cdot \ell_{\mathrm{BC}}(0, f_i)\}, \tag{6}$$

| Reduction | Example instantiation |
|---|---|
| One-versus-all | $\sum_{i \in [L]} \left\{ -y_i \cdot \log \frac{e^{f_i}}{1+e^{f_i}} - (1-y_i) \cdot \log \frac{1}{1+e^{f_i}} \right\}$ |
| Pick-all-labels | $\sum_{i \in [L]} -y_i \cdot \log \frac{e^{f_i}}{\sum_{j \in [L]} e^{f_j}}$ |
| One-versus-all normalised | $\sum_{i \in [L]} \left\{ -\frac{y_i}{\sum_{j \in [L]} y_j} \cdot \log \frac{e^{f_i}}{1+e^{f_i}} - \left(1 - \frac{y_i}{\sum_{j \in [L]} y_j}\right) \cdot \log \frac{1}{1+e^{f_i}} \right\}$ |
| Pick-all-labels normalised | $\sum_{i \in [L]} -\frac{y_i}{\sum_{j \in [L]} y_j} \cdot \log \frac{e^{f_i}}{\sum_{j \in [L]} e^{f_j}}$ |
| Pick-one-label | $-y_{i'} \cdot \log \frac{e^{f_{i'}}}{\sum_{j \in [L]} e^{f_j}}, i' \sim \mathrm{Discrete}\left(\left\{ \frac{y_i}{\sum_{j \in [L]} y_j} \right\}\right)$ |

Table 2: Examples of multilabel losses underpinning various reductions, given labels $y \in \{0,1\}^L$ and predictions $f \in \mathbb{R}^L$. We assume a sigmoid or softmax cross-entropy for the relevant base losses $\ell_{\mathrm{BC}}, \ell_{\mathrm{MC}}$.

where $\ell_{\mathrm{BC}} \colon \{0,1\} \times \mathbb{R} \to \mathbb{R}_+$ is some binary classification loss (e.g., logistic loss). In words, we convert each $(x, y)$ into a positive example for each label with $y_i = 1$, and a negative example for each label with $y_i = 0$. This is also known as the *binary relevance* model [Brinker et al., 2006, Tsoumakas and Vlahavas, 2007, Dembczyński et al., 2010].

**Pick-all-labels** (PAL). Another natural approach involves a multiclass rather than binary loss: we convert each $(x, y)$ for $y \in \mathcal{Y}$ into multi-class observations $\{(x, i) \colon i \in [L], y_i = 1\}$, with one observation per positive label [Reddi et al., 2019]. This can be interpreted as using the multilabel loss

$$\ell_{\mathrm{PAL}}(y, f) \doteq \sum_{i \in [L]} y_i \cdot \ell_{\mathrm{MC}}(i, f), \qquad (7)$$

where $\ell_{\mathrm{MC}} \colon L \times \mathbb{R}^L \to \mathbb{R}_+$ is some multiclass loss (e.g., softmax with cross-entropy or BOWL, per §2.2). While the base loss is multiclass – which, inherently, assumes there is only one relevant label – we compute the *sum* of many such multiclass losses, one for each positive label in $y$.[3] Observe that each loss in the sum involves the entire vector of scores $f$; this is in contrast to the OVA loss, wherein each loss in the sum only depends on the scores for the $i$th label. Further, note that when $L$ is large, one may design efficient stochastic approximations to such a loss [Reddi et al., 2019].

**One-versus-all normalised** (OVA-N). A natural variant of the approach adopted in the above two reductions is to *normalise* the contribution of each loss by the number of positive labels. For the OVA method, rather than independently model each label $y_i$, we thus model *normalised* labels:

$$\ell_{\mathrm{OVA-N}}(y, f) \doteq \sum_{i \in [L]} \left\{ \frac{y_i}{\sum_{j \in [L]} y_j} \cdot \ell_{\mathrm{BC}}(1, f_i) + \left(1 - \frac{y_i}{\sum_{j \in [L]} y_j}\right) \cdot \ell_{\mathrm{BC}}(0, f_i) \right\}. \qquad (8)$$

To gain some intuition for this loss, take the special case of square loss, $\ell_{\mathrm{BC}}(y_i, f_i) = (y_i - f_i)^2$. One may verify that $\ell_{\mathrm{OVA-N}}(y, f) = \sum_{i \in [L]} (y_i' - f_i)^2$ plus a constant, for $y_i' \doteq \frac{y_i}{\sum y_j}$. Thus, the loss encourages $f_i$ to estimate the "normalised labels" $y_i'$, rather than the raw labels $y_i$ as in OVA.

**Pick-all-labels normalised** (PAL-N). Similar to the OVA-N method, we can normalise PAL to:

$$\ell_{\mathrm{PAL-N}}(y, f) \doteq \sum_{i \in [L]} \frac{y_i}{\sum_{j \in [L]} y_j} \cdot \ell_{\mathrm{MC}}(i, f) = \frac{1}{\sum_{j \in [L]} y_j} \cdot \ell_{\mathrm{PAL}}(y, f). \qquad (9)$$

Such a reduction appears to be folk-knowledge amongst practitioners (in particular, being allowed by popular libraries [Abadi et al., 2016]), but as far as we are aware, has not been previously studied. To gain some intuition for this loss, observe that $y_i' \doteq \frac{y_i}{\sum_{j \in [L]} y_j}$ forms a distribution over the labels.

Suppose our scores $f \in \mathbb{R}^L$ are converted to a probability distribution via a suitable link function $\sigma$ (e.g., the softmax), and we apply the log-loss as our $\ell_{\mathrm{MC}}$. Then, $\ell_{\mathrm{PAL-N}}$ corresponds to minimising the cross-entropy between the *true* and *model* distributions over labels:

$$\ell_{\mathrm{PAL-N}}(y, f) = \sum_{i \in [L]} -y_i' \cdot \log \sigma(f_i) = \mathrm{KL}(y' \| \sigma(f)) + \mathrm{Constant}.$$

**Pick-one-label** (POL). In this reduction, given an example $(x, y)$, we select a single random positive label from $y$ as the true label for $x$ [Jernite et al., 2017, Joulin et al., 2017]. This can be understood as a stochastic version of (9), which considers a *weighted* combination of *all* positive labels.

## 4.2 Multilabel reductions: population risks

Each of the above reductions is intuitively plausible. For example, the OVA reduction is a classical approach, which has seen success in multiclass to binary contexts; it is natural to consider its use in multilabel contexts. On the other hand, the PAL reduction explicitly encourages "competition" amongst the various labels if, e.g., used with a softmax cross-entropy loss. Finally, the normalised reductions intuitively prevent instances with many relevant labels from dominating our modelling.

In order to make such intuitions precise, a more careful analysis is needed. To do so, we consider what underlying *multilabel risk* (i.e., expected loss) each reduction implicitly optimises.

**Proposition 5.** *Given a scorer $f : \mathcal{X} \to \mathbb{R}$, the multilabel risks for each of the above reductions are:*

$$R_{\mathrm{OVA}}(f) = \sum_{i \in [L]} \mathbb{E}_{(x,y_i)} [\ell_{\mathrm{BC}}(y_i, f_i(x))] \qquad R_{\mathrm{OVA-N}}(f) = \sum_{i \in [L]} \mathbb{E}_{(x,y_i')} [\ell_{\mathrm{BC}}(y_i', f_i(x))]$$

$$R_{\mathrm{PAL}}(f) = \mathbb{E}_{(x,z)} [N(x) \cdot \ell_{\mathrm{MC}}(z, f(x))] \qquad R_{\mathrm{PAL-N}}(f) = R_{\mathrm{POL}}(f) = \mathbb{E}_{(x,z')} [\ell_{\mathrm{MC}}(z', f(x))],$$

*where $\mathbb{P}(y_i' = 1 \mid x)$ is per (5), and we have defined discrete random variables $z, z'$ over $[L]$ by*

$$\mathbb{P}(z' = i \mid x) \doteq \mathbb{P}(y_i' = 1 \mid x) \tag{10}$$

$$\mathbb{P}(z = i \mid x) \doteq N(x)^{-1} \cdot \mathbb{P}(y_i = 1 \mid x) \tag{11}$$

$$N(x) \doteq \sum_{i \in [L]} \mathbb{P}(y_i = 1 \mid x). \tag{12}$$

Proposition 5 explicates that, as expected, the OVA and OVA-N methods decompose into sums of binary classification risks, while the other reductions decompose into sums of multiclass risks. There are three more interesting implications. First, normalisation has a non-trivial effect: for both OVA and PAL, their normalised counterparts involve modified binary and multiclass label distributions respectively. In particular, while PAL involves $\mathbb{P}(z \mid x)$ constructed from the marginal label probabilities, PAL-N involves $\mathbb{P}(z' \mid x)$ constructed from the "transformed" probabilities in (5).

Second, PAL yields a *weighted* multiclass risk, where the weight $N(x)$ is the *expected number of relevant labels* for $x$. Since PAL treats the multilabel problem as a series of multiclass problems for each positive label, instances with many relevant labels have a greater contribution to the loss. The weight can also be seen as normalising the marginal label probabilities $[\mathbb{P}(y_i = 1 \mid x)]_{i \in [L]}$ to a valid multiclass distribution over the $L$ labels. By contrast, the risk in PAL-N is *unweighted*, despite the losses for each being related by a scaling factor per (9). Intuitively, this is a consequence of the fact that the normaliser can vary across draws from $\mathbb{P}(y \mid x)$, i.e.,

$$\mathbb{E}_{y|x} [\ell_{\mathrm{PAL-N}}(y, f(x))] = \mathbb{E}_{y|x} \left[ \frac{1}{\sum_{j \in [L]} y_j} \cdot \ell_{\mathrm{PAL}}(y, f(x)) \right] \neq \mathbb{E}_{y|x} \left[ \frac{1}{\sum_{j \in [L]} y_j} \right] \cdot \mathbb{E}_{y|x} [\ell_{\mathrm{PAL}}(y, f(x))] .$$

Third, there is a subtle distinction between PAL and OVA. The former allows for the use of an arbitrary multiclass loss $\ell_{\mathrm{MC}}$; in the simplest case, for a base binary classification loss $\ell_{\mathrm{BC}}$, we may choose a loss which treats the given label as a positive, and all other labels as negative: $\ell_{\mathrm{MC}}(i, f) = \ell_{\mathrm{BC}}(1, f_i) + \sum_{j \neq i} \ell_{\mathrm{BC}}(0, f_j)$. This is a *multiclass* version of the one-versus-all reduction [Rifkin and Klautau, 2004]. However, even with this choice, the PAL and OVA risks do *not* agree.

**Lemma 6.** *The risk of the PAL reduction using the multiclass one-verus-all loss is*

$$R_{\mathrm{PAL}}(f) = R_{\mathrm{OVA}}(f) + \sum_{i \in [L]} \mathbb{E}_x [(N(x) - 1) \cdot \ell_{\mathrm{BC}}(0, f_i(x))] .$$

To understand this intuitively, the PAL loss decomposes into one term for each relevant label; for *each* such term, we apply the loss $\ell_{\mathrm{MC}}$, which considers all other labels to be negative. Consequently, every irrelevant label is counted multiple times, which manifests in the extra weighting term above. We will shortly see how this influences the optimal scorers for the reduction.

# 5 Optimal scorers for multilabel reductions

Having explicated the risks underpinning each of the reductions, we are now in a position to answer the question of what precisely they are optimising. We show that in fact each reduction is consistent for *either* the precision@$k$ or recall@$k$; following §3, they cannot be optimal for *both* in general.

## 5.1 Multilabel reductions: Bayes-optimal predictions

We begin our analysis by computing the Bayes-optimal scorers for each reduction. Doing so requires we commit to a particular family of loss functions $\ell_{\mathrm{BC}}$ and $\ell_{\mathrm{MC}}$. We consider the family of *strictly proper losses* [Savage, 1971, Buja et al., 2005, Reid and Williamson, 2010], whose Bayes-optimal solution in a binary or multiclass setting is the underlying class-probability. Canonical examples are the cross-entropy and square loss. In the following, we shall assume our scorers are of the form $f \colon \mathcal{X} \to [0,1]$, so that they output valid probabilities rather than arbitrary real numbers; in practice this is typically achieved by coupling a scorer with a link function, e.g., the softmax or sigmoid.

**Corollary 7.** *Suppose $\ell_{\mathrm{BC}}$ and $\ell_{\mathrm{MC}}$ are strictly proper losses, and we use scorers $f \colon \mathcal{X} \to [0,1]$. Then, for every $x \in \mathcal{X}$ and $i \in [L]$, the Bayes-optimal $f_i^*(x)$ for the OVA and PAL reductions are*

$$f_{\mathrm{OVA},i}^*(x) = \mathbb{P}(y_i = 1 \mid x)$$
$$f_{\mathrm{PAL},i}^*(x) = N(x)^{-1} \cdot \mathbb{P}(y_i = 1 \mid x),$$

*while the Bayes-optimal $f_i^*(x)$ for each of the "normalised" reductions (OVA-N, PAL-N, POL) are*

$$f_i^*(x) = \mathbb{P}(y_i' = 1 \mid x).$$

In words, the unnormalised reductions result in solutions that preserve the ordering of the marginal probabilities, while the normalised reductions result in solutions that preserve the ordering of the "transformed" marginal probabilities. Recalling Corollary 2 and 4, we see that the unnormalised reductions implicitly optimise for *precision*, while the normalised reductions implicitly optimise for *recall*. While this fact is well known for OVA [Dembczyński et al., 2010, Wydmuch et al., 2018], to our knowledge, the question of Bayes-optimality has not been explored for the other reductions.

## 5.2 Multilabel reductions: consistency

Corollary 7 shows that the various reductions' asymptotic targets coincide with those for precision or recall. But what can be said about their *consistency* (in the sense of Equation 4)? For the multiclass reductions, we now show this follows owing to a stronger version of Corollary 7: PAL and PAL-N have *identical* risks (upto scaling and translation) to the precision and recall@$k$, respectively.

**Corollary 8.** *Suppose $\ell_{\mathrm{MC}}$ is the top-$k$ loss in* (2). *Then, using this loss with PAL and PAL-N,*

$$R_{\mathrm{PAL}}(f) = \mathrm{Constant} - k \cdot \mathrm{Prec@}k(f)$$
$$R_{\mathrm{PAL-N}}(f) = \mathrm{Constant} - \mathrm{Rec@}k(f).$$

Despite the prior use of these reductions, the above connection has, to our knowledge, not been noted hitherto. It explicates that two superficially similar reductions optimise for fundamentally different quantities, and their usage should be motivated by which of these is useful for a particular application.

Building on this, we now show that when used with surrogate losses that are *consistent* for top-$k$ error (cf. (4)), the reductions are consistent with respect to the precision and recall, respectively. In the following, denote the regret of a scorer $f \colon \mathcal{X} \to \mathbb{R}^L$ with respect to a multilabel loss $\ell_{\mathrm{ML}}$ by

$$\mathrm{reg}(f; \ell_{\mathrm{ML}}) \doteq \mathop{\mathbb{E}}_{(x,y)} \left[ \ell_{\mathrm{ML}}(y, f(x)) \right] - \inf_{g \colon \mathcal{X} \to \mathbb{R}^L} \mathop{\mathbb{E}}_{(x,y)} \left[ \ell_{\mathrm{ML}}(y, g(x)) \right].$$

We similarly denote the regret for the precision@$k$ and recall@$k$ by $\mathrm{reg}(f; \mathrm{P@}k)$ and $\mathrm{reg}(f; \mathrm{R@}k)$.

**Proposition 9.** *Suppose $\ell_{\mathrm{MC}}$ is consistent for the top-$k$ error. For any sequence $(f_n)_{n=1}^\infty$ of scorers,*

$$\mathrm{reg}(f_n; \ell_{\mathrm{PAL}}) \to 0 \implies \mathrm{reg}(f_n; \mathrm{P@}k) \to 0$$
$$\mathrm{reg}(f_n; \ell_{\mathrm{PAL-N}}) \to 0 \implies \mathrm{reg}(f_n; \mathrm{R@}k) \to 0.$$

Our final analysis is regarding the OVA-N method, which does not have a risk equivalence to the recall@$k$.[4] Nonetheless, its Bayes-optimal scorers were seen to coincide with that of the recall; as a consequence, similar to the consistency analysis of OVA in Wydmuch et al. [2018], we may show that accurate estimation of the transformed probabilities $\mathbb{P}(y_i' = 1 \mid x)$ implies good recall performance. This consequently implies consistency of OVA-N, as the latter can guarantee good probability estimates when equipped with a *strongly proper* loss [Agarwal, 2014], such as for example the cross-entropy or square loss.

**Proposition 10.** *Suppose $\ell_{\mathrm{BC}}$ is a $\lambda$-strongly proper loss. For any scorer $f \colon \mathcal{X} \to [0,1]$,*

$$\mathrm{reg}(f; \mathrm{R}@k) \leq 2 \cdot \mathbb{E}_x \left[ \max_{i \in [L]} |\mathbb{P}(y_i' = 1 \mid x) - f_i(x)| \right] \leq \sqrt{2/\lambda} \cdot \mathrm{reg}(f; \ell_{\mathrm{OVA-N}}).$$

### 5.3 Implications and further considerations

We conclude our analysis with some implications of the above results.

**OVA versus PAL**. Corollary 7 suggests that for optimising precision (or recall), there is no asymptotic difference between using OVA and PAL (or their normalised counterparts). However, a potential advantage of the PAL approach is that it allows for use of tight surrogates to the top-$k$ loss (2), wherein only the *top-$k$* scoring negatives are considered in the loss. In settings where $L$ is large, one can efficiently optimise such a loss via the *stochastic negative mining* approach of Reddi et al. [2019].

**Interpreting model scores**. One subtle implication of Corollary 7 is that for all methods but OVA, the learned scores do *not* reflect the marginal label probabilities. In particular, while the optimal scorer for the OVA and PAL both preserve the order of the marginal probabilities, the latter additionally involves an *instance-dependent* scaling by $N(x)$. Consider, then, an extreme scenario where for some $x \in \mathcal{X}$, *all* labels have marginal $\mathbb{P}(y_i = 1 \mid x) = 1$. Under the OVA reduction, we would assign each label a score of $1$, indicating "perfect relevance". However, under the PAL reduction, we would assign them a score of $\frac{1}{L}$, which would naïvely indicate "low relevance".

For PAL, one can rectify this by positing a form $g_i(x)$ for $\mathbb{P}(y_i = 1 \mid x)$, e.g., $g_i(x) = \sigma(w_i^{\mathrm{T}} x)$. Then, if one uses $f_i(x) = \frac{g_i(x)}{\sum_{j \in [L]} g_j(x)}$, the learned $g_i$'s will model the marginals, upto scaling.

**Learning from multiclass samples**. Our focus in the paper has been on settings where we have access to multilabel samples $(x, y)$, and choose to convert them to suitable binary or multiclass samples, e.g., $(x, z)$ for PAL. A natural question is whether it is possible to learn when we only have *partial* knowledge of the true multilabel vector $y$. For example, in an information retrieval setup, we would ideally like to observe the multilabel vector of *all* relevant documents for a query. However, in practice, we may only observe a *single* relevant document (e.g., the first document clicked by a user issuing a query), which is randomly sampled according to the marginal label distribution (11).

In the notation of Proposition 5, such a setting corresponds to observing multiclass samples from $\mathbb{P}(x, z)$ directly, rather than multilabel samples from $\mathbb{P}(x, y)$. A natural thought is to then minimise the top-$k$ risk (2) directly on such samples, in hopes of optimising for precision. Surprisingly, this does *not* correspond to optimising for precision *or* recall, as we now explicate.

**Lemma 11.** *Pick any $\mathbb{P}(x, y)$ over $\mathcal{X} \times \{0,1\}^L$, inducing a distribution $\mathbb{P}(x, z)$ as in (12). Then,*

$$R_{\mathrm{top}-k}(f) = \mathbb{P}(z \notin \mathrm{Top}_k(f(x))) = \mathbb{E}_x \left[ \sum_{i \notin \mathrm{Top}_k(f(x))} N(x)^{-1} \cdot \mathbb{P}(y_i = 1 \mid x) \right].$$

This risk above is similar to the precision@$k$, *except* for the $N(x)^{-1}$ term. Proposition 5 reveals a crucial missing ingredient that explains this lack of equivalence: when learning from $(x, z)$, one needs to *weight* the samples by $N(x)$, the number of relevant labels. This is achieved implicitly by the PAL reduction, since the loss involves a term for each relevant label. One may further contrast the above to the POL reduction: while this reduction does not weigh samples, it samples labels according to the *transformed* distribution (10), rather than the marginal distribution (11). This distinction is crucial in ensuring that the POL matches the recall@$k$.

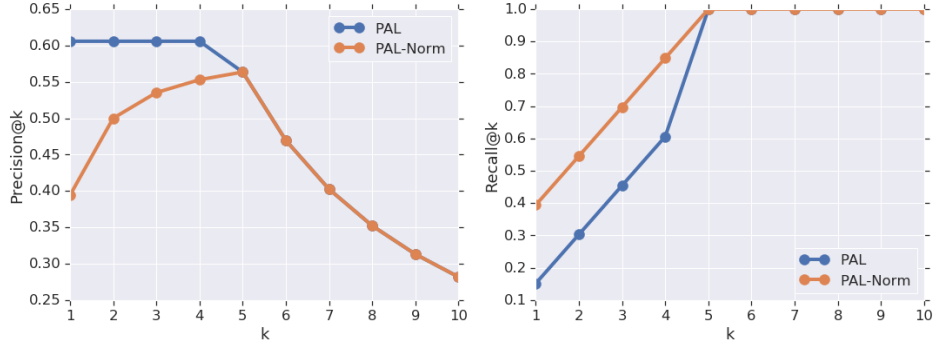

Figure 1: Precision@$k$ and recall@$k$ for the PAL and PAL-N reductions. As predicted by the theory, the former yields superior precision@$k$ performance, while the latter yields superior recall@$k$ performance. The OVA and OVA-N reductions show the same trend, and are omitted for clarity. The maximal number of labels for any instance is $k = 5$, which is the point at which both methods have overlapping curves.

In sum, when we only observe a single relevant label for an instance, care is needed to understand what distribution this label is sampled from, and whether additional instance weighting is necessary.

## 6 Experimental validation

We now present empirical results validating the preceding theory. Specifically, we illustrate that using the pick-all-labels (PAL) reduction versus its normalised counterpart can have significant differences in terms of precision and recall performance. Since the reductions studied here are all extant in the literature, our aim is not to propose one of them as being "best" for multilabel classification; rather, we wish to verify that they simply optimise for different quantities.

To remove potential confounding factors, we construct a synthetic dataset where we have complete control over the data distribution. Our construction is inspired by an information retrieval scenario, wherein there are two distinct groups of queries (e.g., issued by different user bases), which have a largely disjoint set of relevant documents. Within a group, queries may be either generic or specific, i.e., have many or few matching documents. Formally, we set $\mathcal{X} = \mathbb{R}^2$ and $L = 10$ labels. We draw instances from a equal-weighted mixture of two Gaussians, where the Gaussians are centered at $(1, 1)$ and $(-1, -1)$ respectively. For the first mixture component, we draw labels according to $\mathbb{P}(y \mid x)$ which is uniform over two possible $y$: either $y = (\mathbf{1}_{K-1}, 0, \mathbf{0}_K)$, or $y = (\mathbf{0}_{K-1}, 1, \mathbf{0}_K)$, where $K \doteq L/2$. For the second mixture component, the $y$'s are supported on the "swapped" labels $y = (\mathbf{0}_K, 0, \mathbf{1}_{K-1})$, and $y = (\mathbf{0}_K, 1, \mathbf{0}_{K-1})$.

With this setup, we generate a training sample of $10^4$ (instance, label) pairs. We train the OVA, PAL, OVA-N and PAL-N methods, and compute their precision and recall on a test sample of $10^3$ (instance, label) pairs. We use a linear model for our scorer $f$, and the softmax cross-entropy loss for $\ell_{\mathrm{MC}}$.

Figure 1 shows the precision and recall@$k$ curves as $k$ is varied. As predicted by the theory, there is a significant gap in performance between the two methods on each metric; e.g., the PAL method performs significantly better than its normalised counterpart in terms of precision. This illustrates the importance of choosing the correct reduction based on the ultimate performance measure of interest.

## 7 Conclusion and future work

We have studied five commonly used multilabel reductions in a unified framework, explicating the underlying multilabel loss each of them optimises. We then showed that each reduction is provably consistent with respect to *either* precision or recall, but *not* both. Further, we established that the Bayes-optimal scorers for the precision and recall only coincide in special cases (e.g., when the labels are conditionally independent), and so *no* reduction can be optimal for both. We empirically validated that normalised loss functions can yield recall gains over unnormalised counterparts. Consistency analysis for other multilabel metrics and generalisation analysis are natural directions for future work.

## Footnotes

[1] Without this assumption, one may take the convention that $0/0 = 1$ in defining the recall@$k$.

[2] Similar metrics may be defined when $L = 1$ [Kar et al., 2014, 2015, Liu et al., 2016, Tasche, 2018].

[3]This is to be contrast with the *label powerset* approach [Boutell et al., 2004], which treats each distinct label vector as a separate class, and thus creates a multiclass problem with $2^L$ classes.

[4]Consistency of OVA for precision@$k$ was done in Wydmuch et al. [2018].

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
