[Supplementary Material]

# Supplementary material for "Multilabel reductions: what is my loss optimising?"

## A    Proof of results in body

*Proof of Lemma 1.* By definition, and linearity of expectation,

$$
\begin{aligned}
\text{Prec@}k(f) &= \mathop{\mathbb{E}}_{x}\mathop{\mathbb{E}}_{y|x}\left[\frac{1}{k}\cdot|\text{rel}(y)\cap\text{Top}_k(f(x))|\right]\\
&= \mathop{\mathbb{E}}_{x}\mathop{\mathbb{E}}_{y|x}\left[\sum_{i\in[L]}\frac{1}{k}\cdot y_i\cdot[\![i\in\text{Top}_k(f(x))]\!]\right]\\
&= \mathop{\mathbb{E}}_{x}\left[\sum_{i\in\text{Top}_k(f(x))}\frac{1}{k}\cdot\mathop{\mathbb{E}}_{y|x}[y_i]\right]\\
&= \mathop{\mathbb{E}}_{x}\left[\sum_{i\in\text{Top}_k(f(x))}\frac{1}{k}\cdot\mathbb{P}(y_i=1\mid x)\right].
\end{aligned}
\tag{12}
$$

$\square$

*Proof of Lemma 3.* By definition, and linearity of expectation,

$$
\begin{aligned}
\text{Rec@}k(f) &= \mathop{\mathbb{E}}_{x}\mathop{\mathbb{E}}_{y|x}\left[\frac{|\text{rel}(y)\cap\text{Top}_k(f(x))|}{|\text{rel}(y)|}\right]\\
&= \mathop{\mathbb{E}}_{x}\mathop{\mathbb{E}}_{y|x}\left[\frac{\sum_{i\in[L]}y_i\cdot[\![i\in\text{Top}_k(f(x))]\!]}{\sum_{j\in[L]}y_j}\right]\\
&= \mathop{\mathbb{E}}_{x}\mathop{\mathbb{E}}_{y|x}\left[\sum_{i\in\text{Top}_k(f(x))}\frac{y_i}{\sum_{j\in[L]}y_j}\right]\\
&= \mathop{\mathbb{E}}_{x}\left[\sum_{i\in\text{Top}_k(f(x))}\mathop{\mathbb{E}}_{y|x}\left[\frac{y_i}{\sum_{j\in[L]}y_j}\right]\right].
\end{aligned}
\tag{13}
$$

Let us now observe that[5]

$$
\begin{aligned}
\mathop{\mathbb{E}}_{y|x}\left[\frac{y_i}{\sum_{j\in[L]}y_j}\right] &= \mathop{\mathbb{E}}_{y_i}\mathop{\mathbb{E}}_{y_{\neg i}|x,y_i}\left[\frac{y_i}{\sum_{j\in[L]}y_j}\right]\\
&= \mathbb{P}(y_i=1\mid x)\cdot\mathop{\mathbb{E}}_{y_{\neg i}|x,y_i=1}\left[\frac{1}{1+\sum_{j\neq i}y_j}\right]+\\
&\quad (1-\mathbb{P}(y_i=1\mid x))\cdot\mathop{\mathbb{E}}_{y_{\neg i}|x,y_i=0}\left[\frac{0}{0+\sum_{j\neq i}y_j}\right]\\
&= \mathbb{P}(y_i=1\mid x)\cdot\mathop{\mathbb{E}}_{y_{\neg i}|x,y_i=1}\left[\frac{1}{1+\sum_{j\neq i}y_j}\right].
\end{aligned}
\tag{14}
$$

The result follows. $\square$

*Proof of Corollary 4.* The fact that the Bayes-optimal scorers have the stated form follows identically to the proof of the optimal scorers for the precision@$k$ from Wydmuch et al. [2018], except that we now use the probabilities $\mathbb{P}(y_i'=1\mid x)$ in place of the marginals.

We now demonstrate that the ordering of the two probabilities may be different. In fact, the proof of this result is implicit in Wydmuch et al. [2018], who showed the inconsistency of the pick-one-label reduction for precision; we explicate it here. Consider the case of $L = 3$, and a distribution concentrated on a single $x \in \mathcal{X}$, and just two possible $y$:

$$\mathbb{P}(y = (1,1,0) \mid x) = p$$
$$\mathbb{P}(y = (0,0,1) \mid x) = 1 - p,$$

for some $p \in \left(\frac{1}{2}, 1\right)$. Clearly, the marginal probabilities are $(p, p, 1-p)$, and so the first two labels have higher marginal relevance.

Observe now that

$$\mathbb{P}(y'_i = 1 \mid x) = \mathbb{E}_{y|x}\left[\frac{y_i}{\sum_{j \in [L]} y_j}\right] = \begin{cases} p \cdot \frac{1}{2} & \text{if } i = 1 \\ p \cdot \frac{1}{2} & \text{if } i = 2 \\ (1-p) \cdot \frac{1}{1} & \text{if } i = 3. \end{cases}$$

Now suppose $p \in \left(\frac{1}{2}, \frac{2}{3}\right)$. Then, we have $1 - p > \frac{p}{2}$. Consequently, the ordering of the two probabilities will not be the same. In particular, the Bayes-optimal predictor for the recall@1 will favour the label $y_3$, even though it has lowest marginal probability. $\square$

*Proof of Proposition 5.* We provide proofs for each of the reductions in turn.

**OVA**. For the one-versus-all reduction in Equation 6, observe that for a given $x \in \mathcal{X}$ with $f(x) \in \mathbb{R}^L$,

$$\mathbb{E}_{y|x}\left[\ell_{\text{OVA}}(y, f(x))\right] = \mathbb{E}_{y|x}\left[\sum_{i \in [L]} \ell_{\text{BC}}(y_i, f_i(x))\right]$$
$$= \sum_{i \in [L]} \mathbb{E}_{y|x}\left[\ell_{\text{BC}}(y_i, f_i(x))\right]$$
$$= \sum_{i \in [L]} \mathbb{E}_{y_i|x}\left[\ell_{\text{BC}}(y_i, f_i(x))\right].$$

Consequently,

$$R_{\text{OVA}}(f) = \sum_{i \in [L]} \mathbb{E}_{(x,y_i)}\left[\ell_{\text{BC}}(y_i, f_i(x))\right].$$

**PAL**. For the pick-all-labels reduction, observe that for a given $x \in \mathcal{X}$ with $f(x) \in \mathbb{R}^L$, the loss in Equation 7 has expectation

$$\mathbb{E}_{y|x}\left[\ell_{\text{ML}}(y, f(x))\right] = \sum_{i \in [L]} \mathbb{E}_{y|x}\left[y_i \cdot \ell_{\text{MC}}(i, f(x))\right]$$
$$= \sum_{i \in [L]} \mathbb{P}(y_i = 1 \mid x) \cdot \ell_{\text{MC}}(i, f(x))$$
$$= N(x) \cdot \mathbb{E}_{z|x} \ell_{\text{MC}}(z, f(x)),$$

where $N(x) \doteq \sum_{i \in [L]} \mathbb{P}(y_i = 1 \mid x)$ is a normaliser, being the expected number of labels per instance. Here, $z$ is a discrete random variable taking values in $[L]$, with $\mathbb{P}(z = i \mid x) \doteq \frac{\mathbb{P}(y_i=1|x)}{N(x)}$. Consequently,

$$R_{\text{PAL}}(f) = \mathbb{E}_{(x,z)}\left[N(x) \cdot \ell_{\text{MC}}(z, f(x))\right].$$

**OVA-N**. For the normalised one-versus-all reduction in Equation 8, observe that

$$\mathbb{E}_{y|x}\left[\ell_{\text{ML}}(y, f(x))\right] = \sum_{i \in [L]} \mathbb{E}_{y|x}\left[\frac{y_i}{\sum_{j \in [L]} y_j} \cdot \ell_{\text{BC}}(1, f_i(x)) + \left(1 - \frac{y_i}{\sum_{j \in [L]} y_j}\right) \cdot \ell_{\text{BC}}(0, f_i(x))\right]$$
$$= \sum_{i \in [L]} \mathbb{P}(y'_i = 1 \mid x) \cdot \ell_{\text{BC}}(1, f_i(x)) + (1 - \mathbb{P}(y'_i = 1 \mid x)) \cdot \ell_{\text{BC}}(0, f_i(x))$$

$$= \sum_{i \in [L]} \mathbb{E}_{y_i'|x} \left[ \ell_{\mathrm{BC}}(y_i', f_i(x)) \right],$$

where in the third line we used the definition of $\mathbb{P}(y_i' = 1 \mid x)$ (Equation 5) and Equation 14. Consequently,

$$R_{\mathrm{OVA-N}}(f) = \sum_{i \in [L]} \mathbb{E}_{(x,y_i')} \left[ \ell_{\mathrm{BC}}(y_i', f_i(x)) \right].$$

**PAL-N**. For the normalised pick-all-labels reduction observe that for a given $x \in \mathcal{X}$ with $f(x) \in \mathbb{R}^L$, the loss in Equation 9 has expectation

$$\begin{aligned}
\mathbb{E}_{y|x} \left[ \ell_{\mathrm{ML}}(y, f(x)) \right] &= \sum_{i \in [L]} \mathbb{E}_{y|x} \left[ \frac{y_i}{\sum_{j \in [L]} y_j} \cdot \ell_{\mathrm{MC}}(i, f(x)) \right] \\
&= \sum_{i \in [L]} \mathbb{P}(y_i' = 1 \mid x) \cdot \ell_{\mathrm{MC}}(i, f(x)) \\
&= \sum_{i \in [L]} \mathbb{P}(z' = i \mid x) \cdot \ell_{\mathrm{MC}}(i, f(x)) \\
&= \mathbb{E}_{z'} \ell_{\mathrm{MC}}(z', f(x)),
\end{aligned}$$

where in the second line we used the fact that $\mathbb{P}(z' = i \mid x) \doteq \mathbb{P}(y_i' = 1 \mid x)$ is a valid multiclass distribution, as previously noted in Wydmuch et al. [2018] and evident from Equation 14. Consequently,

$$R_{\mathrm{PAL-N}}(f) = \mathbb{E}_{(x,z')} \left[ \ell_{\mathrm{MC}}(z', f(x)) \right].$$

**POL**. The result here trivially follows from the fact that POL is a stochastic version of PAL. $\qquad \square$

*Proof of Lemma 6.* By Proposition 5, we have

$$R_{\mathrm{PAL}}(f) = \mathbb{E}_{(x,z)} \left[ N(x) \cdot \ell_{\mathrm{MC}}(z, f(x)) \right]$$

$$= \mathbb{E}_x \left[ \sum_{i \in [L]} N(x) \cdot \mathbb{P}(z = i \mid x) \cdot \ell_{\mathrm{MC}}(i, f(x)) \right]$$

$$= \mathbb{E}_x \left[ \sum_{i \in [L]} \mathbb{P}(y_i = 1 \mid x) \cdot \left\{ \ell_{\mathrm{BC}}(1, f_i(x)) + \sum_{j \neq i} \ell_{\mathrm{BC}}(0, f_j(x)) \right\} \right]$$

$$= \mathbb{E}_x \left[ \sum_{i \in [L]} \mathbb{P}(y_i = 1 \mid x) \cdot \left\{ \ell_{\mathrm{BC}}(1, f_i(x)) - \ell_{\mathrm{BC}}(0, f_i(x)) + \sum_{j \in [L]} \ell_{\mathrm{BC}}(0, f_j(x)) \right\} \right]$$

$$= \mathbb{E}_x \left[ \sum_{i \in [L]} \mathbb{P}(y_i = 1 \mid x) \cdot \left\{ \ell_{\mathrm{BC}}(1, f_i(x)) - \ell_{\mathrm{BC}}(0, f_i(x)) \right\} \right] +$$

$$\mathbb{E}_x \left[ \left( \sum_{i \in [L]} \mathbb{P}(y_i = 1 \mid x) \right) \cdot \left( \sum_{j \in [L]} \ell_{\mathrm{BC}}(0, f_j(x)) \right) \right]$$

$$= \mathbb{E}_x \left[ \sum_{i \in [L]} \mathbb{P}(y_i = 1 \mid x) \cdot \left\{ \ell_{\mathrm{BC}}(1, f_i(x)) - \ell_{\mathrm{BC}}(0, f_i(x)) \right\} \right] +$$

$$\mathbb{E}_x \left[ N(x) \cdot \left( \sum_{j \in [L]} \ell_{\mathrm{BC}}(0, f_j(x)) \right) \right]$$

$$
= \mathop{\mathbb{E}}_{x} \left[ \sum_{i \in [L]} \mathbb{P}(y_i = 1 \mid x) \cdot \ell_{\mathrm{BC}}(1, f_i(x)) + \mathbb{P}(y_i = 0 \mid x) \cdot \ell_{\mathrm{BC}}(0, f_i(x)) - \ell_{\mathrm{BC}}(0, f_i(x)) \right] +
$$

$$
\mathop{\mathbb{E}}_{x} \left[ N(x) \cdot \left( \sum_{j \in [L]} \ell_{\mathrm{BC}}(0, f_j(x)) \right) \right]
$$

$$
= \sum_{i \in [L]} \mathop{\mathbb{E}}_{x, y_i} \left[ \ell_{\mathrm{BC}}(y_i, f_i(x)) \right] + \mathop{\mathbb{E}}_{x} \left[ (N(x) - 1) \cdot \left( \sum_{j \in [L]} \ell_{\mathrm{BC}}(0, f_j(x)) \right) \right].
$$

$\square$

*Proof of Corollary 7.* We provide proofs for each of the reductions in turn.

**OVA**. For the one-versus-all reduction, we may compute each $f_i^*(x)$ independently. When $\ell_{\mathrm{BC}}$ is a strictly proper loss, these will by definition be equal $\mathbb{P}(y_i = 1 \mid x)$.

**PAL**. For the pick-all-labels reduction, the factor $N(x)$ only modifies the marginal distribution of $x$. Since $N(x) > 0$, this weighting factor will not affect the Bayes-optimal solution. Thus, since $\ell_{\mathrm{MC}}$ is strictly proper, the Bayes-optimal prediction will by definition be $\mathbb{P}(z = i \mid x) = \frac{\mathbb{P}(y_i = 1 \mid x)}{N(x)}$.

**OVA-N**. For the normalised one-versus-all reduction, since $\ell_{\mathrm{BC}}$ is a strictly proper loss, by definition $f_i^*(x) = \mathbb{P}(y_i' = 1 \mid x)$.

**PAL-N**. For the normalised pick-all-labels reduction, since $\ell_{\mathrm{MC}}$ is a strictly proper loss, and we have a multiclass risk, by definition $f_i^*(x) = \mathbb{P}(z' = i \mid x) = \mathbb{P}(y_i' = 1 \mid x)$.

**POL**. The result here trivially follows from the fact that POL is a stochastic version of PAL. $\square$

*Proof of Corollary 8.* Observe first that, by (12),

$$
k \cdot \mathrm{Prec}@k(f) = \sum_{i \in [L]} \mathop{\mathbb{E}}_{(x, y_i)} \left[ y_i \cdot [\![ i \in \mathrm{Top}_k(f(x)) ]\!] \right]
$$

$$
= \sum_{i \in [L]} \mathop{\mathbb{E}}_{(x, y_i)} \left[ y_i \cdot (1 - \ell_{\mathrm{top}-k}(i, f(x))) \right]
$$

$$
= \sum_{i \in [L]} \mathop{\mathbb{E}}_{y_i} [y_i] - \mathop{\mathbb{E}}_{(x, y_i)} \left[ y_i \cdot \ell_{\mathrm{top}-k}(i, f(x)) \right]
$$

$$
= \mathrm{constant} - \sum_{i \in [L]} \mathop{\mathbb{E}}_{(x, y_i)} \left[ y_i \cdot \ell_{\mathrm{top}-k}(i, f(x)) \right]
$$

$$
= \mathrm{constant} - \mathop{\mathbb{E}}_{(x, y)} \left[ \sum_{i \in [L]} y_i \cdot \ell_{\mathrm{top}-k}(i, f(x)) \right]
$$

$$
= \mathrm{constant} - R_{\mathrm{PAL}}(f).
$$

Similarly, by (13),

$$
\mathrm{Rec}@k(f) = \mathop{\mathbb{E}}_{(x, y)} \left[ \sum_{i \in [L]} \frac{y_i}{\sum_{j \in [L]} y_j} \cdot [\![ i \in \mathrm{Top}_k(f(x)) ]\!] \right]
$$

$$
= \mathop{\mathbb{E}}_{(x, y)} \left[ \sum_{i \in [L]} \frac{y_i}{\sum_{j \in [L]} y_j} \cdot (1 - \ell_{\mathrm{top}-k}(i, f(x))) \right]
$$

$$
= \mathop{\mathbb{E}}_{(x, y)} \left[ \sum_{i \in [L]} \frac{y_i}{\sum_{j \in [L]} y_j} \right] - \mathop{\mathbb{E}}_{(x, y)} \left[ \sum_{i \in [L]} \frac{y_i}{\sum_{j \in [L]} y_j} \cdot \ell_{\mathrm{top}-k}(i, f(x)) \right]
$$

$$= \text{constant} - \underset{(x,y)}{\mathbb{E}} \left[ \sum_{i \in [L]} \frac{y_i}{\sum_{j \in [L]} y_j} \cdot \ell_{\text{top}-k}(i, f(x)) \right]$$

$$= \text{constant} - R_{\text{PAL}-\text{N}}(f).$$

$\square$

*Proof of Proposition 9.* For clarity, we will explicate here the dependence of all quantities on the type of risk (multiclass or multilabel reduction), and the base loss. For simplicity, we will assume the existence of a *surrogate regret bound* for the top-$k$ risk: that is, we assume that for every scorer, the surrogate loss $\ell_{\text{MC}}$ satisfies

$$\text{reg}_{\text{MC}}(f; \ell_{\text{top}-k}) \leq \Psi \left( \text{reg}_{\text{MC}}(f; \ell_{\text{MC}}) \right)$$

for some function $\Psi$ such that $\lim_{z \to 0+} \Psi(z) = 0$, where $\text{reg}_{\text{MC}}(f; \ell_{\text{MC}})$ denotes multiclass regret for a scorer using a loss $\ell_{\text{MC}}$. This clearly implies consistency.

We provide bounds for the precision and recall in turn.

**Precision bound**. By Corollary 8,

$$\text{Prec@}k(f) = \text{Constant} - \frac{1}{k} \cdot R_{\text{PAL}}(f; \ell_{\text{top}-k}).$$

Thus,

$$\text{reg}(f; \text{P@}k) = \frac{1}{k} \cdot \text{reg}_{\text{PAL}}(f; \ell_{\text{top}-k})$$

where $\text{reg}_{\text{PAL}}(f; \ell_{\text{MC}}) \doteq R_{\text{PAL}}(f; \ell_{\text{MC}}) - \inf_{g \colon \mathcal{X} \to \mathbb{R}} R_{\text{PAL}}(g; \ell_{\text{MC}})$ is the regret under the PAL reduction, using a base multiclass loss of $\ell_{\text{MC}}$.

By Proposition 5, for any multiclass loss $\ell_{\text{MC}}$,

$$R_{\text{PAL}}^{\mathbb{P}}(f; \ell_{\text{MC}}) = C \cdot \underset{(\bar{x}, z)}{\mathbb{E}} [\ell_{\text{MC}}(z, f(\bar{x}))] = C \cdot R_{\text{MC}}^{\bar{\mathbb{P}}}(f; \ell_{\text{MC}}),$$

where, if $x$ has distribution with density $p(x)$ with respect to base measure $\mu$, $\bar{x}$ has distribution with density $\bar{p}(x) \doteq C^{-1} \cdot p(x) \cdot N(x)$, and $C \doteq \int_{\mathcal{X}} p(x') \cdot N(x') \, \mathrm{d}\mu(x)$. Note that $C \leq L < +\infty$ since $N(x) \in [0, L]$ for every $x \in \mathcal{X}$. Note also that the above explicates the dependence of the risks on the underlying distributions, as they are distinct in the LHS and RHS.

The above implies that the multilabel PAL risk given a distribution $\mathbb{P}(x, z)$ is expressible as a constant times a standard multiclass risk over a distribution $\bar{\mathbb{P}}(\bar{x}, z)$. Thus, a Bayes-optimal scorer for the PAL risk must satisfy

$$f^* \in \underset{f \colon \mathcal{X} \to \mathbb{R}}{\text{argmin}} R_{\text{PAL}}^{\mathbb{P}}(f; \ell_{\text{MC}}) = \underset{f \colon \mathcal{X} \to \mathbb{R}}{\text{argmin}} R_{\text{MC}}^{\bar{\mathbb{P}}}(f; \ell_{\text{MC}}).$$

Consequently, if $f^*$ denotes the Bayes-optimal scorer for the PAL risk, we have

$$\text{reg}_{\text{PAL}}^{\mathbb{P}}(f; \ell_{\text{MC}}) = C \cdot \underset{(\bar{x}, z)}{\mathbb{E}} [\ell_{\text{MC}}(z, f(\bar{x})) - \ell_{\text{MC}}(z, f^*(\bar{x}))]$$

$$= C \cdot \text{reg}_{\text{MC}}^{\bar{\mathbb{P}}}(f; \ell_{\text{MC}}).$$

We have thus reduced the *multilabel* regret on the left-hand side to a *multiclass* regret on the right-hand side. Further, in doing so, we have introduced a distribution $\bar{\mathbb{P}}(\bar{x}, z)$ whose marginal density $\bar{p}$ is *distorted* compared to the original $p$.

Now, since $\ell_{\text{MC}}$ is assumed to be a consistent surrogate to the top-$k$ loss,

$$\text{reg}(f; \text{P@}k) = \frac{1}{k} \cdot \text{reg}_{\text{PAL}}^{\mathbb{P}}(f; \ell_{\text{top}-k})$$

$$= \frac{C}{k} \cdot \text{reg}_{\text{MC}}^{\bar{\mathbb{P}}}(f; \ell_{\text{top}-k})$$

$$\leq \frac{C}{k} \cdot \Psi \left( \text{reg}_{\text{MC}}^{\bar{\mathbb{P}}}(f; \ell_{\text{MC}}) \right)$$

$$= \frac{C}{k} \cdot \Psi \left( \mathrm{reg}_{\mathrm{PAL}}^{\mathbb{P}}(f; \ell_{\mathrm{MC}}) \right),$$

where we have used the fact that a surrogate regret bound holds for *any* distribution, even one where the marginals are distorted. We thus have a surrogate regret bound for the precision@$k$.

**Recall bound**. By Corollary 8,

$$\mathrm{Rec}@k(f) = \mathrm{Constant} - R_{\mathrm{PAL-N}}(f; \ell_{\mathrm{top}-k}).$$

Thus,

$$\mathrm{reg}(f; \mathrm{R}@k) = \mathrm{reg}_{\mathrm{PAL-N}}(f; \ell_{\mathrm{top}-k}).$$

By Proposition 5,

$$R_{\mathrm{PAL-N}}(f; \ell_{\mathrm{top}-k}) = \mathop{\mathbb{E}}_{(x,z')} \left[ \ell_{\mathrm{top}-k}(z', f(x)) \right]$$

where $\mathbb{P}'(x, z')$ is as per (11). That is, the multilabel PAN risk is a multiclass risk. Now, since $\ell_{\mathrm{MC}}$ is assumed to be a consistent surrogate to the top-$k$ loss,

$$\begin{aligned}
\mathrm{reg}(f; \mathrm{R}@k) &= \mathrm{reg}_{\mathrm{PAL-N}}^{\mathbb{P}}(f; \ell_{\mathrm{top-k}}) \\
&= \mathrm{reg}_{\mathrm{MC}}^{\mathbb{P}'}(f; \ell_{\mathrm{top-k}}) \\
&\leq \Psi \left( \mathrm{reg}^{\mathbb{P}'}(f; \ell_{\mathrm{MC}}) \right) \\
&= \Psi \left( \mathrm{reg}_{\mathrm{PAL-N}}^{\mathbb{P}}(f; \ell_{\mathrm{MC}}) \right).
\end{aligned}$$

We thus have a surrogate regret bound for the recall@$k$. $\qquad\square$

*Proof of Proposition 10.* Observe that by Lemma 3 and Corollary 7,

$$\begin{aligned}
\mathrm{reg}(f; \mathrm{R}@k) &= \mathop{\mathbb{E}}_{x} \left[ \sum_{i \in \mathrm{Top}_k(f(x))} \mathbb{P}(y_i' = 1 \mid x) - \sum_{i \in \mathrm{Top}_k(f^*(x))} \mathbb{P}(y_i' = 1 \mid x) \right] \\
&= \mathop{\mathbb{E}}_{x} \left[ \sum_{i \in \mathrm{Top}_k(f(x))} \mathbb{P}(y_i' = 1 \mid x) - \sum_{i \in \mathrm{Top}_k(\mathbb{P}(y'|x))} \mathbb{P}(y_i' = 1 \mid x) \right].
\end{aligned}$$

The first inequality thus follows exactly as per Wydmuch et al. [2018, Theorem 2]: we simply swap $\mathbb{P}(y_i = 1 \mid x)$ with $\mathbb{P}(y_i' = 1 \mid x)$. Thus,

$$\mathrm{reg}(f; \mathrm{R}@k) \leq 2 \cdot \mathop{\mathbb{E}}_{x} \left[ \max_{i \in [L]} |f_i(x) - \mathbb{P}(y_i' = 1 \mid x)| \right]. \tag{15}$$

The second inequality is a standard consequence of existing bounds for strongly proper losses [Agarwal, 2014]. By Proposition 5,

$$\begin{aligned}
R_{\mathrm{OVA-N}}(f) &= \sum_{i \in [L]} \mathop{\mathbb{E}}_{(x,y_i)} \left[ \ell_{\mathrm{BC}}(y_i, f_i(x)) \right] \\
&= \sum_{i \in [L]} R_{\mathrm{BC}}(f_i),
\end{aligned}$$

where $R_{\mathrm{BC}}$ denotes a binary classification risk. Since the risk decomposes across each $i \in [L]$, the Bayes-optimal scorers may be computed separately for $i$. The regret thus similarly decomposes as:

$$\begin{aligned}
\mathrm{reg}(f; \ell_{\mathrm{OVA-N}}) &= \sum_{i \in [L]} \mathrm{reg}(f_i; \ell_{\mathrm{BC}}) \\
&\geq \sum_{i \in [L]} \frac{\lambda}{2} \cdot \mathop{\mathbb{E}}_{x} \left[ (f_i(x) - \mathbb{P}(y_i' = 1 \mid x))^2 \right]
\end{aligned}$$

$$= \frac{\lambda}{2} \cdot \mathbb{E}_x \left[ \sum_{i \in [L]} (f_i(x) - \mathbb{P}(y_i' = 1 \mid x))^2 \right]$$

$$\geq \frac{\lambda}{2} \cdot \mathbb{E}_x \left[ \max_{i \in [L]} (f_i(x) - \mathbb{P}(y_i' = 1 \mid x))^2 \right] \quad \text{by non-negativity of each term}$$

$$= \frac{\lambda}{2} \cdot \mathbb{E}_x \left[ \left( \max_{i \in [L]} |f_i(x) - \mathbb{P}(y_i' = 1 \mid x)| \right)^2 \right]$$

$$\geq \frac{\lambda}{2} \cdot \left( \mathbb{E}_x \left[ \max_{i \in [L]} |f_i(x) - \mathbb{P}(y_i' = 1 \mid x)| \right] \right)^2 \quad \text{by Jensen's inequality}$$

$$\geq \frac{\lambda}{2} \cdot \left( \max_{i \in [L]} \mathbb{E}_x |f_i(x) - \mathbb{P}(y_i' = 1 \mid x)| \right)^2 \quad \text{by Jensen's inequality,}$$

where in the second line, we used the fact that $\ell_{\mathrm{BC}}$ is strongly proper, and Agarwal [2014, Theorem 13]; and in the fifth line, we used the fact that the maximum is over a sequence of $L$ nonnegative quantities, so that the maximum of their squares is the square of their maximum. We thus have

$$\max_{i \in [L]} \mathbb{E}_x |f_i(x) - \mathbb{P}(y_i' = 1 \mid x)| \leq \sqrt{\frac{2}{\lambda}} \cdot \sqrt{\mathrm{reg}(f; \ell_{\mathrm{OVA-N}})}.$$

Applying this to (15), the claim follows. □

*Proof of Lemma 11.* Observe that

$$R_{\mathrm{top}-k}(f) = \mathbb{P}(z \notin \mathrm{Top}_k(f(x)))$$

$$= \mathbb{E}_x \mathbb{E}_{z|x} \llbracket z \notin \mathrm{Top}_k(f(x)) \rrbracket$$

$$= \mathbb{E}_x \mathbb{E}_{z|x} [\ell_{\mathrm{top}-k}(z, f(x))]$$

$$= \mathbb{E}_x \left[ \sum_{i \in [L]} \mathbb{P}(z = i \mid x) \cdot \ell_{\mathrm{top}-k}(i, f(x)) \right]$$

$$= \mathbb{E}_x \left[ \sum_{i \notin \mathrm{Top}_k(f(x))} N(x)^{-1} \cdot \mathbb{P}(y_i = 1 \mid x) \right],$$

where in the last line we used the definition of $\mathbb{P}(z = i \mid x)$ from Equation 11. □

## Footnotes

[5]We have also used here the assumption that $\mathbb{P}(\mathbf{0}_L\mid x)=0$.