[Reviews · NeurIPS 2019]

Reviewer 1



Originality: This is a very original contribution which studies commonly used reductions for multi-label classification. The obtained results are not only surprising, but also very important from the practical point of view. Quality: All the results are presented in a formal way. The claims are clear and theoretically justified. The empirical illustration is rather limited, but this is not the main focus of the paper (the results for OVA and OVA-N should be given in the Appendix). Clarity: The paper is clearly written. Nevertheless the list below contains some comments regarding the text: - P(0_L | x) = 0 => I would prefer to not make such assumption. For example, some benchmark datasets contain examples with no labels. - Prec@k and Rec@k: these measures have certainly been used before Lapin et al. 2018. - Lemma 3 => There are undefined quantities in the lemma and typos in its proof. The final result is correct, but readability should be improved (at least the authors should say what y_{\not i} means and that they use the fact that P(y) = P(y_i)P(y_{\not i}|y_i)). Moreover, P(y_i' = 1) should be given in both forms, i.e. as in Lemma 3 and as E_y|x [ y_i/\sum_j y_j ] (the latter form is used often in the proofs of the other results). - binary relevance: the name binary relevance was used before Dembczynski et al 2010, but not sure where for the first time :( - The analysis could mention the label powerset approach which reduces a multi-label problem to a multi-class problem with exponentially many meta-labels. In this approach, however, we do not get easily marginal quantities. - Eq. \ell_{PAN-N} => I suppose that there should be log before the link function in the sum - The analysis in this paper is methodologically similar to the one from "On label dependence and loss minimization in multi-label classification", MLJ 2012. - Appendix: please proof-read the derivations. Significance: This is a significant contribution containing somehow simple, but surprising results with high practical potential.

Reviewer 2



It is a well-written paper with clear structure, easy to follow. It contains adequate discussions of related work, and the proposed approach is experimentally validated. Regarding novelty, the paper follows the footsteps of Wydmuch et al., 2018, but the extension is substantial, as it covers a much boarder range of reduction techniques and the discussion on recall in Wydmuch et al., 2018 is not explicit. The authors also call for caution in interpreting the produced probabilities scores of the reduction techniques. But isn't it rather trivial? It is not a criticism; I'd just like to point it out in case I've missed something.

Reviewer 3



This paper analyzes a relevant machine learning problem, as it intends to provide additional insights into existing methods that have been introduced in an ad-hoc manner. From that perspective, the paper is interesting. However, it is not easy to understand the main results. I have the feeling that the write-up and design of the paper could be substantially improved. Let me give some specific comments to make this point clear. Up to Section 4 the paper is reasonably clear, but in Section 4 things get a bit messy. I have some problems to understand the losses in Equations 6 and 7, because the l_BC and l_MC are never defined explicitly. Please give formulas here. Many different ways of expressing the logistic and softmax loss exist, so I would like to have it precise here. I can’t see the usefulness of Equation 8. Is this loss reasonable? Let’s take a training example with 100 positive labels. Treating each positive label as positive gets a weight of 1/100, while treating positives as negatives gets a weight of 99/100. So, this seems to suggest that false positives should be heavily penalized. Maybe I am missing here something, but this does not make sense to me. Equation 9 is also a strange variant. Here the denominator in the sum does not depend on i, so it can be moved in front of the sum. As a result, this term just reweighs the importance of an instance, but it does not influence the risk minimizer for that instance. PAL and PAL-N should therefore have the same risk minimizer, but Corollary 7 seems to suggest a different result. I am confused here. Should Corollary 7 not be named Theorem 7? In mathematics a corollary is a direct implication from a theorem. I don’t see from which theorem the corollary would follow here. Traditionally, there are two ways to optimize task-based loss functions in machine learning: (a) optimizing a convex and differentiable approximation of the task loss during training, (b) fitting a probabilistic model at training time, followed by a loss-specific inference step at test time in a decision-theoretic framework. For me, a big point of confusion is that the approaches are somewhat mixed in this paper. Wouldn’t it be easier to analyze the different methods in a classical decision-theoretic framework? In essence this would boil down to using accuracy for l_BC and l_MC. In a nutshell, this is an interesting paper, but I think the write-up could be improved. In general the results are not very clear and counterintuitive. ---- After author rebuttal: ---- my main motivation for giving a somewhat-lower score was some technical things that were not clear to me. I am satisfied with the author's response and will raise my score.

[Author Response · NeurIPS 2019]

Thanks for the detailed feedback! We are glad that reviewers found our results to be interesting.

**R1**:

> $\mathbb{P}(\mathbf{0}_L|x) = 0 \rightarrow$ I would prefer to not make such assumption.

This was done to simplify the definition of recall, since if no labels are relevant ($y = \mathbf{0}_L$), we would naïvely have to
compute $\frac{0}{0}$. We can however remove this assumption, and note that $y = \mathbf{0}_L$ requires fixing a choice for the recall.

> Lemma 3 $\rightarrow$ There are undefined quantities in the lemma and typos in its proof.

We will explicate the meaning of $y_{\neg i}$ (which, as the reviewer correctly inferred, refers to all labels but the $i$th one), and
also add the two forms of $\mathbb{P}(y_i' = 1 \mid x)$ as suggested.

We will incorporate the additional citations and other minor comments, which are appreciated.

**R2**:

> The authors also call for caution in interpreting the produced probabilities scores of the reduction techniques. But
isn't it rather trivial? It is not a criticism; I'd just like to point it out in case I've missed something.

The fact discussed in Section 5.3 that most reductions do not output marginal label probabilities indeed follows
immediately from our results. We simply wished to explicate that while OVA with logistic and PAL with softmax
cross-entropy loss produce probability estimates, precisely *what* these probabilities measure are fundamentally different.

**R3**:

> I have some problems to understand the losses in Equations 6 and 7, because the $\ell_{\mathrm{BC}}$ and $\ell_{\mathrm{MC}}$ are never defined.

We work with abstract binary/multiclass losses $\ell_{\mathrm{BC}}/\ell_{\mathrm{MC}}$ to highlight that our results are not tied to specific choices. We
tried to make the quantities concrete by providing examples of the logistic and softmax cross-entropy loss on Lines 142
and 148. We use their standard definitions: for binary $y_i \in \{0, 1\}$ the logistic loss is $\ell_{\mathrm{BC}}(y_i, f_i) = \log(1 + e^{-(2y_i-1)\cdot f_i})$,
while the softmax cross-entropy loss is as defined on Line 98. We will clarify this in our revision.

> I can't see the usefulness of Equation 8 . . . So, this seems to suggest that false positives should be heavily penalized.

To get some intuition, take the special case of square loss, $\ell_{\mathrm{BC}}(y_i, f_i) = (y_i - f_i)^2$. One may verify that $\ell_{\mathrm{OVA-N}}(y, f) =$
$\sum_{i \in [L]}(y_i' - f_i)^2$ plus a constant, for $y_i' = \frac{y_i}{\sum y_j}$. Thus, the provided weighting scheme encourages $f_i$ to estimate the
"normalised labels" $y_i'$, rather than the raw labels $y_i$. One can obtain similar results for the logistic and hinge loss.

Observe also that the scale of $y_i' \in \{0, \frac{1}{100}\}$ in your example, which is much smaller than that of $y_i \in \{0, 1\}$. To model
this compressed range of values, we thus need to shrink our predictions for the positives closer to 0. Placing a large
weight on the negative term ($\ell_{\mathrm{BC}}(0, f_i)$) when $y_i = 1$ achieves precisely this. We will add a discussion in our revision.

> Equation 9 is also a strange variant. Here the denominator in the sum does not depend on i, so it can be moved in
front of the sum. . . . PAL and PAL-N should therefore have the same risk minimizer.

To get some intuition, per Line 154, the effect of normalisation is to create a valid distribution $y_i'$ over labels. The loss
thus seeks to minimise the discrepancy between $y_i'$ and the model distribution $q_i$ over labels; e.g., for the cross-entropy
loss, we choose $q$ to minimise $-\sum_{i \in [L]} y_i' \cdot \log q_i$, or equally, $\mathrm{KL}(y_i' \| q_i)$.

It is true that $\sum_{j \in [L]} y_j$ can be moved outside the sum. However, it is not true that this is a constant weight in the risk: for

any fixed $x$, we have to compute $\mathbb{E}_{y|x}\left[\frac{1}{\sum_{j \in [L]} y_j} \cdot \sum_{i \in [L]} y_i \cdot \ell_{\mathrm{MC}}(i, f)\right] \neq \frac{1}{\mathbb{E}_{y|x}[\sum_{j \in [L]} y_j]} \cdot \sum_{i \in [L]} \mathbb{E}_{y|x}[y_i \cdot \ell_{\mathrm{MC}}(i, f)]$
in general. Equality only holds when the number of labels is constant across $x$; we will make this point explicit.

> Traditionally, there are two ways to optimize task-based loss functions . . . For me, a big point of confusion is that the
approaches are somewhat mixed in this paper. Wouldn't it be easier to analyze . . . using accuracy for $\ell_{\mathrm{BC}}$ and $\ell_{\mathrm{MC}}$.

Ideally, it is always desirable to directly optimise the downstream task-specific measure of ultimate interest. In multilabel
retrieval settings, these are typically the precision@$k$ and recall@$k$; however, their direct optimisation is challenging.
This has motivated the reductions proposed in prior work, which have been informally motivated as optimising *some*
task-specific multilabel loss. It is precisely the motivation of this work to understand exactly what loss this is.

Both precision@$k$ and recall@$k$ implicitly use the top-$k$ loss (Corollary 8). For $k = 1$ this is exactly the misclassification
loss, which is in line with the reviewer's suggestion about using accuracy for $\ell_{\mathrm{BC}}$ and $\ell_{\mathrm{MC}}$.

[Meta-Review · NeurIPS 2019]

The paper analyzes several multi-label reduction approaches, and show the performance measure related to each approach. The reviewers agree that some of the results are novel, solid and non-trivial, such as how precision@k is optimized by pick-all-labels reduction and recall@k is optimized by pick-one-label reduction. While the empirical results are somewhat limited, the reviewers agree that the theoretical contributions are sufficient to recommend acceptance. Given that the results in the paper are clean and can be easily summarized, spotlight presentation is recommended. The authors are encouraged to improve the clarity of some parts of the writing.